# Using a Social-ecological Regime Shift Approach to Understand the Transition from Livestock to Game Farming in the Eastern Cape, South Africa



**Therezah Achieng [1,*], Kristine Maciejewski [1], Michelle Dyer [2] and Reinette Biggs [1,2]**

[1]  Center for Complex Systems in Transition (CST), School of Public Leadership, Stellenbosch University, Private Bag XI, Matieland 7602, South Africa; krismacski@gmail.com (K.M.); oonsie@spl.sun.ac.za (R.B.)

[2]  Stockholm Resilience Centre, Stockholm University, 106 19 Stockholm, Sweden; michellejdyer@gmail.com

*  Correspondence: terrytrizahachieng@gmail.com; Tel.: +25-4726-772-438

**Abstract:** This study explored the shift in land use from livestock farming to game farming in the Eastern Cape, South Africa, from a social-ecological regime shift perspective. A regime shift can be defined as a large, persistent change in the structure and function of the intertwined social and ecological components of a landscape. This research focused on the Amakhala game reserve as a case study to understand how the shift affected the provision of ecosystem services and human wellbeing. We used remote sensing techniques to quantify changes in vegetation and found evidence of vegetation recovery following the shift. We then conducted interviews with both landowners and farmworkers and used participatory mapping to understand their perceptions of the main drivers and social-ecological impacts of the shift in land use. Social narratives revealed stark differences in different stakeholders' perceptions, highlighting that the change in land use had varied implications for, and were perceived differently by, different stakeholders. Farmworkers emphasized changes in social structures that weakened community bonds and erased valued connections to the land. At the same time, they increased employment of women, skills development, and increased wages as benefits of the new game farming regime. Landowners, on the other hand, indicated financial gains from the land use change. The transition therefore resulted in trade-offs that surfaced as social, economic, and cultural losses and gains. These changes, especially in social relationships and community structures, have implications for resilience and possible future pathways of development in the region.

**Keywords:** regime shift; Amakhala; social-ecological system; land use change; ecosystem services; human wellbeing; trade-offs

## 1. Introduction

Land use change, the transformation in the functional role of land for economic activities [1], has major implications for biodiversity and a wide variety of ecosystem services [2]. Changes in land use can also lead to changes in land cover, i.e., the change in natural cover of a landscape, e.g., from savannah to cropland [3]. It is these changes in land cover that directly impact biodiversity [2,3] as well as a range of ecosystem services produced by human interaction with nature or ecological systems [4], including provisioning (e.g., food and water), regulating (e.g., water purification and control of soil erosion), and cultural services (e.g., recreation and aesthetic values) [5]. These changes in ecosystem services in turn have direct impacts on human wellbeing by affecting a range of resources and processes that underpin people's livelihoods, identities, and cultural practices. The changes may also have social and economic effects on human societies, economies, and health [6–8].

In the Eastern Cape region of South Africa, there has been a marked shift in land use from livestock farming to game farming since the early 1980s [9]. Game farming in the context of South Africa is a conservation-oriented farming practice where wild (game) animals are raised to stock wildlife areas for hunting, ecotourism, and sold for food [10]. Livestock farming on the other hand connotes the acts of animal husbandry, where domesticated animals rely on grass and other forage to produce milk, meat, eggs, and other products. Adoption of game farming in the Eastern Cape province reached a peak in 2000 [11]. This shift was precipitated by a variety of factors that rendered livestock farming less economically viable [12]. Specifically, the growth in game farming was boosted by change in wildlife ownership laws and renewed conservation interests in the 1970s, coupled with the introduction of stock reduction schemes after the prolonged drought of the 1960s, which lowered cattle prices [11]. The shift to game farming has increased, accelerated by political, socio-economic, and ecological factors [11], and has had a variety of impacts, directly impacting biodiversity and ecosystem services, as well as the livelihoods of farm owners and farm workers in the region [13].

As apparent in the Eastern Cape, land use change involves changes in the intertwined ecological, social, and economic functions of the land [14], and can be understood as an emergent feature of social-ecological systems [15]. Social-ecological systems are complex adaptive systems characterized by multi-scale feedback, self-organization, and non-linear dynamics emerging from the interactions of social and ecological factors in a landscape [14,16]. These feedback processes and interactions lead complex systems to organize in certain functional structures that tend to be relatively stable and persistent [16]. However, if the system is exposed to ongoing incremental changes or large shocks that exceed a systemic threshold or tipping point [17], the system can abruptly shift into an alternative structure with a different set of functions [16,17]. Such shifts between contrasting and persistent structures of a complex system are referred to as regime shifts [16], and have been documented in ecosystems, social systems, and social-ecological systems [18,19].

Regime shifts have been increasingly useful as a conceptual framework for understanding the drivers and consequences of large, persistent systemic changes in social-ecological systems [16], and informing preventative and restorative management actions [17]. Understanding regime shifts is not only important due to their potential impacts on human societies and economies [6,20], but also because they are difficult to predict and costly or even impossible to reverse [21]. Changes that lead to regime shifts can involve large external shocks, such as fires or floods, slow changes already present in the system, or a combination of these driving a system towards a tipping point [22,23]. The point at which these changes will trigger a regime shift is often unknown [22], and regime shifts therefore often occur as a surprise [17]. When they occur, regime shifts often have large impacts on ecosystems and the services they generate [24] with consequent implications for human economies, societies, and human wellbeing [8,18]. For example, the shift from grassy to woody savanna in African savannas threatens the provision of ecosystem services, such as grazing for livestock, clean water, and habitat for some herbivores [15].

Regime shifts and their impacts on ecosystem services tend to affect different stakeholders in different ways. Impacts can vary socially across gender, duties and responsibilities [25,26], economically along income brackets, and within different sectors. For instance, although game farming has generated new opportunities and new forms of added value, including eco-tourism, trophy hunting and even game meat production, it is still contested in the Eastern Cape [27]. This is because game farming may deny local communities' sense of space, create dispossession, and the loss of rights of access to land [13]. Between 1994 and 2004, for instance, 2.35 million dwellers in the province were evicted from commercial livestock farms to give way for game farming [28]. This occurred despite land reform programs put in place by the state to secure people's rights of occupancy and access to land, to prevent forceful evictions, and to regulate relations between dwellers and owners. Such outcomes also have their roots in histories of racism, sexism, and capitalism (colonial and recent) in this region. These historical factors have determined land distribution, rights, and negotiating power among landowners

and local communities. The larger socio-economic implications of these trends include loss of residence, unemployment, sprawl of informal settlements in urban areas, and weakened social bonds [29].

This paper applies a social-ecological regime shift framework [16] to understand the transition from livestock farming to game farming, focusing specifically on the Amakhala game reserve in the Eastern Cape of South Africa. While various studies have investigated different dimensions of this land use transition, none have applied a regime shift approach to understand the interconnected social and ecological drivers, impacts, and implications for human wellbeing. The study specifically sought to address (i) how has landcover changed before, during, and after the shift to game farming, (ii) how have ecosystem services and benefits that farmworkers obtain from local ecosystems changed, and (iii) how have landowners and farmworkers respectively experienced the change?

## 2. Materials and Methods

The study employed a combination of qualitative and quantitative methods, including social narratives and remote sensing techniques. Different stakeholders were engaged to understand how perspectives and experiences varied between different users, specifically between farmworkers and landowners.

### 2.1. Study Area

Amakhala game reserve is a private game farm located in the Eastern Cape province in South Africa, at 33°26′45.07″ S and 26°7′24.05″ E [30], as shown in Figure 1, which comprises a number of farms that were previously used for livestock farming. The total area of the reserve is 8500 ha, and the dominant ecosystem types include thicket and savannah. The thicket biome consists of thorny scrub forests mixed with grasslands, especially in high-lying areas [31]. The savannah biome consists of grass and shrub-trees, mainly used for grazing animals, and previously for cattle/livestock production [32]. Prior to converting into game farms, the land use in Amakhala game reserve was predominantly agrarian, including chicory and maize farming, livestock farming, and different forms of subsistence farming. Large portions of land were used for commercial livestock farming, mostly stocked with sheep and goat.

Amakhala game reserve is owned by a total of eight landowners who represent the original landowners of the smaller agriculture and livestock farms that now make up the game reserve. According to the landowners, the decision to convert from livestock to game farming was primarily influenced by a six-year drought that occurred between 1989 and 1995. This drought affected livestock grazing and reduced water quality and quantity, making the livestock sector less economically viable [11,12]. The amalgamation of these neighboring livestock farms created the Amakhala Conservation Centre in 1999, which covers over 8500 ha [30].

With the conversion into a game farm, game species were introduced including cheetah, buffalo, elephant [33], and different types of antelope. Old houses were converted into lodges and Amakhala became an ecotourism operating game reserve, with eight lodges and camping facilities. Establishment of each of these lodges commenced in the 2000s [12]. Services offered by these private reserves have since attracted a number of domestic and international tourists [34,35]. In 2018, the Eastern Cape was ranked as the third most visited province in South Africa [34]. Ecotourism can potentially stimulate economic growth and alleviate poverty, especially in marginalized areas [36].

Amakhala presents an interesting case study to assess both the ecological and social impacts of the shift to game farming over time. Prior to the conversion to game farming, farmworkers and farm dwellers were able to move about the land and formed a community; that is, they interacted and shared residential areas, and social and cultural activity spaces on the livestock farms. However, these interactions changed with the erection of fences around the game farms, and the relocation of farm workers to nearby towns and centers. The nature of social relations changed and the concept of a 'community' on the game farms took on very different characteristics.

In this study, we focused on three of the original farms that comprise Amakhala: Woodbury, Leeuwenbosch, and Carnarvon Dale, as shown in Figure 1. We interviewed both game farm landowners, as well as farmworkers, to explore differences in their perspectives and experiences of the shift from livestock to game farming. Some of the farmworkers that participated in the study were from the original livestock farms before they were amalgamated and converted into game farms. This provided the opportunity to gain a more holistic perspective of the change by drawing on different types of knowledge [37].

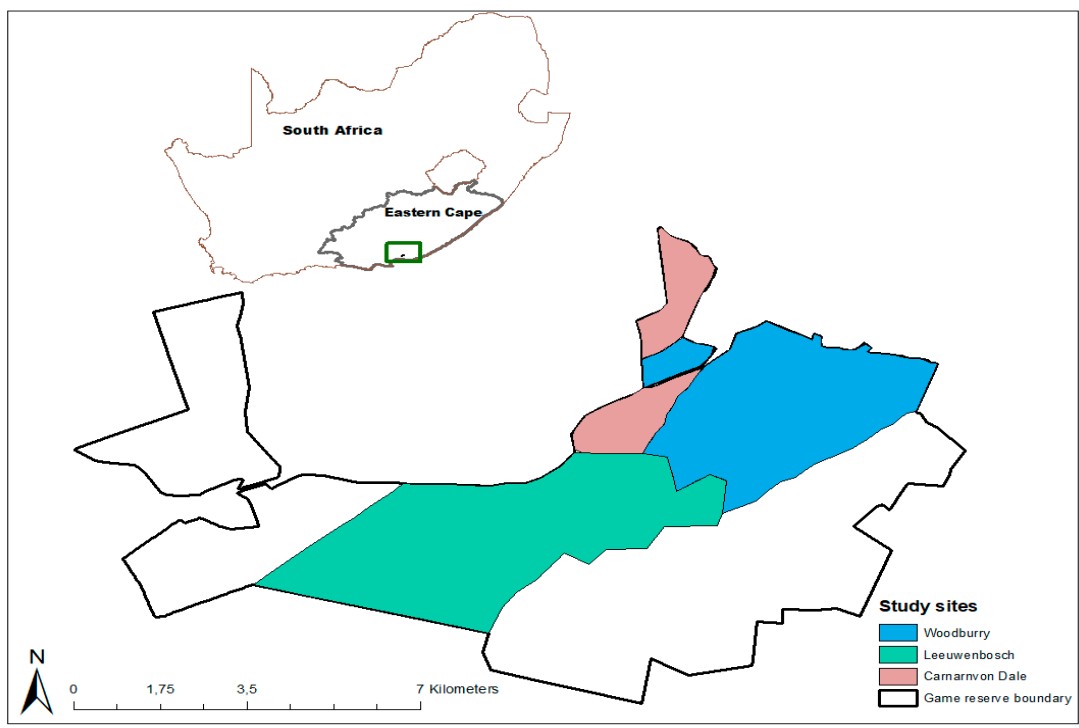

**Figure 1.** Location of the three study sites in Amakhala game reserve, in the Eastern Cape Province of South Africa (ArcGIS 10.5.1).

## 2.2. Research Methodology

This study used both quantitative and qualitative research approaches to provide a broader perspective of the changes in this social-ecological system. Quantitative methods included geographic information systems (GIS) and remote sensing (RS) time series analysis, which were used to detect and analyze land cover changes over time. Qualitative methods included literature analysis, participatory mapping, focus group discussions, stakeholder interviews/narratives, and qualitative modelling, which described the social perceptions of land use change over time.

### 2.2.1. Geographic Information System and Remote Sensing (GIS/RS)

Medium resolution images from Landsat 5–8, specifically for the years 1984, 1992, 2009, and 2017, were used to assess changes in vegetation cover. The specific years were chosen due to image suitability; images with limited cloud cover, which were within the month of May of each year. The specific years and months were selected to ensure consistency and minimize the effect of seasonal variations. For the years 1984 and 1992, Landsat 5 spectral bands were used. Landsat 7 spectral bands were used for the year 2009, while Landsat 8 bands were acquired for the most recent year, 2017. Geometric correction was not necessary since all the images were acquired from USGS (LIT), where data is already corrected using ground control points (GCPs) [38].

Nine land cover classes were classified from the images based on the Land Cover Classification System 1.8.3 (LCCS) [39]. These classes included rangeland, bare areas, built up areas, thicket,

cultivated fields, riparian vegetation, wetland, newly revegetated, and 'others'. The 'others' category included those land cover classes that could not be categorized as one of the other eight classes. Newly revegetated referred to vegetated areas covering areas identified as bare areas in the early Landsat images. The newly revegetated class was herein classified as a separate group in order to see where change had happened. The built up areas land class in this case was defined as any form of building that that existed, emerged, or changed in the temporal space, including but not limited to settlements, guest houses, or small shopping centers within the delineated study area. A regression analysis was used to test for statistical significance of the landcover change over time. Significance was assigned when $p < 0.01$.

### 2.2.2. Participatory Mapping and Focus Group Discussions with Farmworkers

Participatory mapping was used to understand farmworkers' perceptions of the change from livestock to game farming. Participatory mapping is a social mapping tool used for capturing stakeholders' perspectives and represents spatial knowledge of stakeholders through sketching [40]. It is used to understand the historical and present relationships of people with the environment they live in and derive their livelihood from, and understand the ways in which communities connect with their environment/landscapes [41]. While land cover change over time can be assessed through remote sensing techniques [42], participatory mapping by stakeholders complemented this method by capturing perceptions linked to changes witnessed in the landscape.

A total of ten farmworkers, two men and eight women between the ages of 20 and 70 years, participated in the exercises at the three study sites (5 at Woodbury, 3 at Leeuwenbosch, and 2 at Carnarvon Dale), between 13th to 14th May 2018. Farmworkers were selected on condition that: (i) they currently worked on the game farm and; (ii) they had previously worked on the livestock farms before they were converted into game farms. The selected participants were representative of the farmworkers at the three selected sites in terms of age, gender, and type of work. More women in the sample size implied that there were more women employed on the game farms compared to livestock farms as a benefit and cost of the transition, later explained in the discussion. The farmworkers were guided through a participatory mapping exercise in groups of between 2 and 5 people. Each group was asked to sketch the historical landscape as far back as they could remember, and then draft a sketch of the current landscape. Participants were presented with flip charts and asked to sketch different features in relation to one another, for instance, location of rivers and roads in relation to houses and churches, location of cemeteries in relation to dwellings, and so on. Then, on the other flip charts, they were asked to sketch what the landscape looked like now under the game farm as a land use. This process was followed by focus group discussions, where participants in their respective groups were asked to describe how they used various aspects of sketched features in the landscape, and the consequences of the shift in these features (from livestock to game farming) for how they used the landscape. Changes in ecosystem services and valued social and community features related to the sketches from this exercise were used to summarize Table 2. The focus group discussions offered spaces in which participants could share their emotions and memories linked to certain benefits lost or acquired over time. Each group was also asked to discuss their perceptions of the key drivers of change and how this affected their wellbeing. Wellbeing is herein defined as living conditions, social functioning, economic status, and cultural conditions of farm dwellers in both regimes. The aspects of wellbeing defined in this case was linked to direct provision ecosystem services as food, water, firewood, and game meat, and valued social and community features including churches, schools, and cemeteries. Responses were noted and recorded with participants' consent.

### 2.2.3. Interviews with Landowners

Semi-structured interviews were conducted with the landowners to understand their perceptions of the main drivers and impacts of the livestock–game farm transition. The use of interviews plays a fundamental role in capturing and understanding people's deeper emotions and thoughts relating to

their space, a place they considered, or used to consider, home [41]. It also captures and contextualizes ideas, stories, and reflections relating to the broader social-ecological system in which people are embedded. Four landowners and one game reserve manager were individually interviewed. The interviews covered certain aspects of the interviewee's demography, including age and gender and history of the farms.

## 3. Results

### 3.1. Land Cover Analysis

From 1984 to 2009, Amakhala game reserve was mostly comprised of bare areas, with a maximum extent of 62% bare areas recorded in 1992, as shown in Figure 2. By 2017, bare areas had declined to 20% of the total area, with newly revegetated areas increasing to 26% of the total area. By 2017, there had been an increase in thicket (14%), and built up areas (16%), while the area devoted to rangeland decreased (14%), and crop fields had disappeared. The change in percentage land cover type between 1984 and 1992 was statistically (F(1.7) = 41.4, $p < 0.03$, $R^2 = 0.86$). There was, however, no significant change in land cover between 1984 and 2017 (F(1.7) = 0.91, $p < 0.30$, $R^2 = 0.11$).

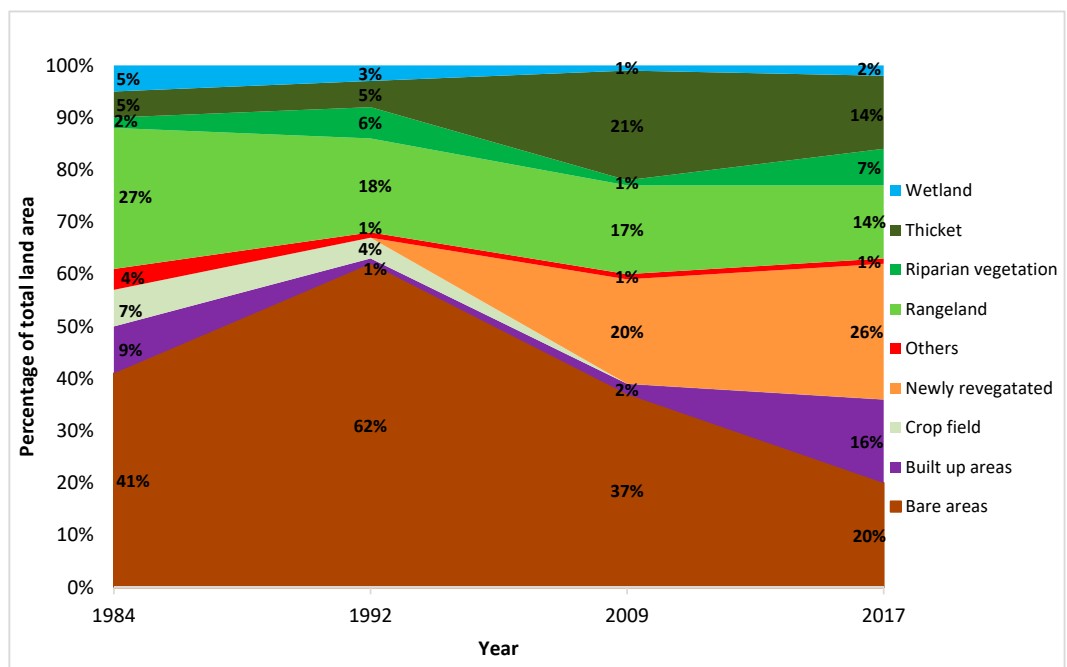

**Figure 2.** The change in land cover of Amakhala game reserve from 1984 to 2017.

### 3.2. Impacts of Land Use Change on Provision of Ecosystem Services and Human Wellbeing

Based on the participatory mapping and focus group discussions, it is evident that there were substantial differences in the benefits (provisioning ecosystem services as well as valued social and community features) provided by livestock farms compared to game farms, across all three sites, as shown in Table 1. The shift from livestock farms to game farms led to a decrease in the provision of food from livestock and gardens, wood fuel, and game meat. Spiritual practices however remained consistent in both land uses, although churches were moved to different locations within the game farms.

A common finding across all three sites was that when the livestock farms were converted into game farms, most farmworkers, along with their families, were relocated to surrounding locations such as Paterson and other nearby towns in the Eastern Cape. This movement away from the farms altered the social networks among farmworkers and personal connections linked to the land as a place referred to as home. Concomitantly, the nature of ecosystem service provision changed in terms of quality and quantity, and accessibility and availability.

**Table 1.** A summary of changes in ecosystem services and valued social and community features provided by the livestock versus game farm regimes.

| Site | Livestock Regime | | Game Farm Regime | |
|---|---|---|---|---|
| | Provisioning Services | Valued Social and Community Features | Provisioning Services | Valued Social and Community Features |
| **Woodbury** | 1. Food crops<br>2. Vegetable gardens<br>3. Wood<br>4. Ration<br>5. Tapped water<br>6. Livestock (pigs, goats, and chickens)<br>7. Plough for themselves | 1. Community games<br>2. Church<br>3. Cultural sports | 1. Chicken<br>2. Tapped water | 1. Church moved closer<br>2. Abandoned residential houses on the farm |
| **Leeuwenbosch** | 1. Game meat<br>2. Ration<br>3. Wood<br>4. River water<br>5. Chicory and maize<br>6. Livestock (pigs, cows, and chickens) | 1. Cemetery<br>2. Church | 1. Limited wood for fuel<br>2. Limited water for vegetables<br>3. Less livestock and chickens<br>4. Piped water | 1. Methodist church<br>2. Gatherings for celebration<br>3. Traditional celebrations |
| **Carnarvon Dale** | 1. Wood<br>2. Ration<br>3. Livestock (chickens, goats, and cows)<br>4. Dam water<br>5. River water<br>6. Game meat<br>7. Beans, pumpkin, and other vegetables | 1. Cultural activities and ceremonies<br>2. Methodist church<br>3. Social interaction with other workers<br>4. Community concerts and drama shows<br>5. Bazaars | 1. Water from 'jojo' tanks. | 1. Church<br>2. Limited social gatherings |

In the livestock regime, people felt a strong connection to nature, which was expressed through a sense of place and place-based identity. The place provided intrinsic and intimate values, including connection with loved ones, both living and those that had been lost, through visiting cemeteries located on the farms near residences. For example, a woman from the Leeuwenbosch site explained: *"I cried when the cemetery was fenced as part of the reserve. I had just lost a close family member and even though there was another cemetery started near the village where we lived, the bodies still remained buried in the soil and we could not get them out. We had to stop visiting the graves because we were no longer allowed in."* (Focus group discussion, 13 May 2018).

Day-to-day activities on the livestock farms also enabled a strong connection to the place and amongst each other. Working on the livestock farms and forming social groups with fellow workers and their families led to social gatherings, cultural events, and barter trading within the community. These gatherings strengthened social networks, important for social learning and personal acknowledgement. This provided a context to reconcile internal conflicts and share experiences with one another. For example, a woman from the Carnarvon Dale site narrated: *"We used to work as a family. We forged intimate connections based on trust that allowed us to plan community celebrations, dances, and exchange goods. Since there was trust among us, there was certainty in the quality of goods we exchanged among us, and if faulty in any case, it would be rectified."* (Focus group discussion, 13 May 2018).

The conversion from livestock farming to game farming affected people's sense of physical and livelihood security, as they no longer felt free to move around. Access to provisioning ecosystem services, such as water from the rivers and wood from the forests, was suddenly restricted. The introduction of large game also instilled a sense of fear and also led to the loss of freely available food including game meat. For instance, a man from the Woodbury site recalled: *"We used to hunt for game meat (kudu, imbabala, ihodi, incanda etc.) freely to supplement food sources which was mostly from the farm and our little vegetable gardens. We also fetched firewood from the bushes used for cooking at home and during braai or community get together. There is no more hunting because there are lions, cheetahs, elephants, and other dangerous ones. There are imbabala (impalas), but we cannot hunt because it is restricted. We also have to buy wood now if we want to braai or cook using woodfuel."* (Focus group discussion, 13 May 2018).

With the shift to game farming, social networks and relationships between farmworkers were altered. New configurations that have emerged through game farming became the economic nodes around which everything revolved. While some participants lamented the social changes and loss of community on livestock farms, this was balanced with the need to earn money and the employment opportunities offered by the game farms. Increased wages were especially seen as beneficial because

of the responsibility to support their families. For example, a man from the Woodbury site stated: "*The money I used to get paid when I was milking cows was something like R295 because I did not need to buy much, but I still needed to take care of my children. So, it was not enough, but now, I get enough money to fully support my family.*" (Focus group discussion, 13 May 2018).

With the shift to game farms, farmworkers had access to better infrastructure, education and healthcare facilities, and mobility was easier, while the existence of childcare facilities ensured their young ones were taken care of while they were at work. However, this altered social relationships. For example, a woman from the Leeuwenbosch site responded: "*There is a creche nearby where we take our kids before coming to work, that helps because we do not need to employ a house help. But we spend very less time with them because we are always working. We also don't get to meet as a community as we used to in the farms.*" (Focus group discussion, 13 May 2018).

To the farmworkers, the erection of game fences demarcated boundaries, and most importantly, secured the wild animals in the game reserve. The livestock farms were more accessible to communities compared to game farms, which have more impermeable fencing and are dangerous to access due to the types of animals they contain. As interpreted by farmworkers, the animals were caged in while the community of people, previously resident, were caged out.

In summary, the livestock farms were associated with more accessibility and stronger social bonds, but low wages (mostly paid to male employees), while the game farms were associated with more economic benefits, but a weakening of the intimate space considered a 'community'. Game farms also facilitated the employment of more women and skill building among staff, which generated a sense of pride absent in the livestock regime. Rather than one regime being clearly more beneficial or desirable than another, each was associated with a different set of perceived costs and benefits, as shown in Table 2.

**Table 2.** Comparison of the key benefits and costs of livestock farming compared to game farming in the Amakhala game reserve, as perceived by farmworkers and landowners.

| Livestock Regime | Game Farm Regime |
| --- | --- |
| • Space for living and community interactions | • Sense of pride in the job |
| • Physical and livelihood security | • More women employed |
| • Healthier and cheaper food | • Good infrastructure |
| • Sense of community | • More schools |
| • Strong family ties | • Skill building among staff |
| • Low wages | • Higher wages |
| • Skewed gender employment (mostly men) | • Biodiversity conservation |

*3.3. Perceived Drivers of the Land Use Change across Farmworkers and Landowners*

Perceptions of the main drivers of the change from livestock to game farming differed between landowners and farmworkers. The farmworkers believed that the main factor influencing farmers' decision to shift into game farming was profit driven, along with the fear of losing land to communities through expropriation as part of the ongoing national process of land reform in South Africa. It was largely felt that the change to game farming was a 'selfish' act, related to individual farmers' 'greed', and fear of sharing or losing their land, which reinforced their desire to adopt game farming.

In contrast, the landowners felt that the decision to shift from livestock farming to game farming centered around drought and the conservation benefits of game farming. The prolonged drought from 1989 to 1995 led to the drying up of the Bushman's River, along with other water sources that sustained livestock on the farms. The quality and quantity of cultivated and natural pasture for livestock was also reduced by the drought, impacting the quality and quantity of livestock products. The resulting economic pressures influenced the farmers decision to convert their pastoral land into game farms.

A second important factor in driving the change in land use was that landowners saw game farming practice as a sustainable land use that increases the nation's protected area estate and contributes to

the conservation of biodiversity. The incremental adoption of game farming catalyzed the spread of ideas and motivations to conserve biodiversity by farmers. The success of neighboring farms, such as Shamwari Game Reserve, and their perceived increase in profit generation, made game farming increasingly attractive to surrounding neighbors. As conversion gradually spread among livestock farmers in the area, the idea of an increased tourism industry in the Eastern Cape was strengthened, and thus reinforced the possibility of landowners to shift to game farming.

## 4. Discussion

The transition from livestock farming to game farming in the Amakhala area had profound ecological as well as social consequences, which we argue represent a social-ecological regime shift. Three criteria for identifying potential social-ecological regime shifts have been identified [16–19]: (1) a large change or reorganization of a social-ecological system has been observed or proposed; (2) the change affects the set of ecosystem services provided by the system, with potential consequences for human wellbeing; and (3) established or proposed feedback mechanisms exist that create and maintain the different regimes so that the change is persistent and not readily reversible. We discuss the shift from livestock to game farming in Amakhala in light of these criteria, emphasizing that the shift is associated with a complex set of inter-related social and ecological changes that lead to a complex set of changes in costs and benefits from an ecological, social, and economic perspective.

*Ecological Changes and Impacts*: The land cover analysis showed clear and persistent changes, with the revegetation of bare areas, and a decrease in rangeland and cultivated areas, following the conversion from livestock to game farming. The drought from 1989 to 1995 in South Africa [11], which dried up Bushman's River in the Eastern Cape, likely explains the very high proportion (62%) of bare cover in Amakhala in 1992, which represented a significant increase over the eight years from 1984. The decline in bare areas in later years, ten years after most farmers converted their land to game farms, suggests a recovery in the vegetation cover, which was likely facilitated by the end of the drought in 1996. Higher rainfall would have increased soil moisture and facilitated regeneration [43], reducing disturbance of the soil from grazing and other livestock-related activities. The shift from livestock to game farming therefore had a positive impact in terms of the extent of natural vegetation and biodiversity values.

*Social Changes and Impacts*: The establishment of game farms radically altered the networks of social relations attached to the farms comprising the Amakhala game reserve, a finding similar to other studies in the Eastern Cape [13]. From the farmworkers' perspectives, the livestock farms were seen as their homes, a place that allowed social networks to be established, which was integral to their sense of community. Following the conversion to game farms, the farmworkers were moved off the game farms in towns, which meant they had to commute to work each day. The workers now living in rental housing in towns and nearby centers have different communities compared to what previously existed in the villages on the livestock farms. Farmworkers who lived on the livestock farms had homes and certain benefits absent in the rental housing after they relocated, including a relatively good life (growing their own food) and cheap cost of living [10]. The workers felt they lost valued social networks that strengthened community bonds, and that they lost their homes and access to the land, through erection of fences. Winter et al. allude to this nature of interaction as relationships or elements that play key roles in building resilience of social-ecological systems by self-organizing and maintaining the balance of the system in the event of a disturbance [44]. Reorganizations in the social structures and networks that connected communities through meaningful bonds is akin to what Winter et al. allude to as biocultural elements, seen as social-ecological keystones, which cannot be substituted without compromising the structure and function of the system [44]. As a result, caging through inaccessible fencing gave rise to a 'metaphoric caging', where day-to-day relationships, social circles, and family bonds revolved around the game farm as a privatized industry. For the farmworkers, these changes in the social fabric of their communities were generally perceived as negative.

*Economic Changes and Impacts*: Game farms have been seen as land uses with more economic activity [1] and gains generated from tourist activities [28]. This study confirmed that the economic benefits contributed to individual financial wellbeing and may extend beyond the reserve. Over and above the perspectives that emerged, the increase in profit generated from the game farm sector contributes to the GDP of the province [36], a benefit that according to a provincial report, extends beyond the scale of the reserve and may be counted at a national scale [34,35]. The economic gains were important to both landowners and farmworkers and contributed to their financial wellbeing, and were perceived as a positive impact of the shift.

*Gender Impacts*: Perceptions of the impacts of the land use change appeared to be differentiated by gender, duties and responsibilities of farmworkers, their relationships with the landowners, and also the number of years they worked on the farm. The change in economic status was very apparent among women workers. While women's incomes may have increased, their socially prescribed roles as family carers and household managers did not change. This aligns with studies of the gender of paid and unpaid labor worldwide and across socio-economic categories—when women take on paid work, their unpaid labor (caring for families) does not decrease, resulting in an increase in their overall labor demands [25,26]. In this case, women testified that their domestic duties did not decrease, even though their formal work duties in the game farm regime increased. At the same time, the provisioning and valued social and community features provided by the livestock regime, that supported their unpaid labor roles, were greatly reduced. This suggests that the shift impacted women more significantly than men. While men's paid work duties may have changed with the regime change, there was no indication that they took on previously female gendered roles in the household even though women took on additional paid work, along with their usual unpaid duties. Therefore, while the shift contributed to economic agency for women, it resulted in an increase in their overall paid and unpaid workload.

*Established Feedback that locks the regime in place*: Game farming is currently seen by landowners as a unique land use involving the establishment of fences and water sources for purchased wild animals, as well as significant investment in the establishment of lodges and training of staff. These large investments are likely to lock this regime in place, together with the profit generated by tourists visiting and staying in the reserve, which generates employment for farmworkers, and salary for landowners. This new regime also involves a new management structure, the growing positive economic reputation of the game farming sector in the province, and a shift in the identity of individual farmers from livestock farming to a wildlife/tourism operation. The shift from livestock to game farming in the Eastern Cape is ongoing, and the insights from this study highlight the differentiated impacts which could help inform policies to reduce the social and economic costs of this change.

Based on the significant and sustained impacts that the shift from livestock to game farming has had on the functioning of ecosystems and human wellbeing of the local community, and the feedback that locks the new game farming regime in place, this research concludes that the shift from livestock to game farming is indeed a social-ecological regime shift. This shift is associated with a complex set of changes in the costs and benefits accruing to different groups and subgroups affected by the change, with no clearly preferred regime [45], as summarized in Table 2.

## 5. Conclusions

A regime shift perspective can help provide a holistic view of the interconnected changes in a social-ecological system. In this case, it provided a deeper, integrated view of the livestock and game farm regimes, which unpacked often overlooked—but crucial—social, economic, and cultural aspects of the transition. By viewing the two land use types as two regimes, we were able to study the interconnected social and ecological features of each regime and unpack the total set of costs and benefits associated with landscape change. Key highlights of this research included: (i) how the shift between the two regimes may yield new and differentially beneficial outcomes depending on the social positioning and perspective of stakeholders, and (ii) the importance of using a systems approach when looking at land use change over time in order to understand the various interconnected costs and

benefits of such shifts. This understanding can be used in resource management to manage differently from a relational perspective and forge socially informed and context-specific policies.

A key limitation of a regime shift approach in the context of social-ecological systems, however, is in establishing tipping points. The social variables that play primary roles in driving a regime shift or maintaining a certain regime often interact in complex ways and cannot be quantified to establish clear thresholds of change that trigger the shift from one regime to another [23]. Another hurdle in using this lens in a social-ecological context is in reconciling different narratives or interpretations of the same event. However, these differences can help illuminate the complex ways in which change is experienced and impacts on different groups, and can be augmented with historical and other sources to gain a deeper understanding of a particular change.

This study highlights that there are not only substantial ecological and social impacts associated with the shift from livestock to game farming, but that the interactions and relationships in the system are also strongly affected—which may in turn influence the resilience of the system, i.e., the capacity of the system to deal with further change and disturbance. Stakeholders' perception of the land use changes indicated strong mismatches in perceived social outcomes, representing differentiated trade-offs associated with the shift. Although some trade-offs may appear to cancel out, the amount of loss or gain that has been incurred as a result of the regime shift cannot be framed as a simple "trading" exercise due to the qualitative nature of social relations. We indicate this as an important area for further research building on these findings.

**Author Contributions:** Conceptualization, T.A. and K.M.; methodology, M.D.; software, M.D.; validation, T.A., M.D., and K.M.; formal analysis, T.A.; investigation, T.A.; resources, R.B.; data curation, T.A.; writing—original draft preparation, T.A.; writing—review and editing, R.B.; visualization, T.A.; supervision, K.M., M.D., and R.B.; project administration, K.M. and R.B.; funding acquisition, R.B. All authors have read and agreed to the published version of the manuscript.

**Funding:** This research was funded by TRECCAFRICA INTRA-ACP Mobility programme and the DST/NRF South African Research Chairs Initiative (SARChI), grant 98766 held by Prof Reinette (Oonsie) Biggs. Kristine Maciejewski, Michelle Dyer, and Reinette Biggs were also supported by the Guidance for Resilience in the Anthropocene: Investments for Development (GRAID) project funded by the Swedish International Development Agency (Sida) hosted at Stockholm Resilience Centre, Sweden, and Reinette Biggs is also supported by a Young Researchers Grant from the Vetenskapsrådet in Sweden (grant 621-2014-5137).

**Acknowledgments:** Data acquired for this research would have never been possible without the consent and participation of the game reserve manager and landowners in Amakhala game reserve, South Africa. I very much appreciate farm workers of the reserve with whom I had the privilege to engage in discussions. Many thanks to the two translators who passionately dedicated their time to help engage with farm workers to explore their narratives.

**Conflicts of Interest:** The authors declare no conflict of interest.

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
