# Peer review of "Using a Social-ecological Regime Shift Approach to Understand the Transition from Livestock to Game Farming in the Eastern Cape, South Africa"

_land, doi:10.3390/land9040097_

Round 1

Reviewer 1 Report

This is a novel paper, and I appreciate the direction that the researchers have taken in their inquires.  I only have a few critiques, which are articulated below.

  • The authors have basically self-cited their own work in the realm of social-ecological regime shifts. This is not a good practice when trying to demonstrate the validity of your work.  These researchers are not the only ones who have looked at regime shifts in social-ecological systems.  They need to explain how their work either supports or provides contrary evidence to the work of researchers other than themselves in this area. Consider using the conclusions of the below study to assess the validity of this study's findings:
    • Winter, K.B.; Lincoln, N.K.; Berkes, F. The Social-Ecological Keystone Concept: A Quantifiable Metaphor for Understanding the Structure, Function, and Resilience of a Biocultural System. Sustainability 201810, 3294.
  • Table 2 is a bit confusing. How is "low wages" a perceived benefit?  It seems like that is more of a con than a pro.  To more clearly present the community's perspectives on the two approaches, there should be a presentation of both perceived pros as well as cons for each scenario.
  • Regime shifts are transitional periods between stable states. The term "stable state" is not used at all in the manuscript, which is problematic.  Is there a preferred stable state from the community's perspective?  This study could make a stronger point about how management can either induce a regime shift towards a preferred stable state or induce a regime shift away from a preferred stable state.  It's more likely that the regime shift when towards a preferred stable state for one group in the community, and away from a preferred stable state for another group.  These insights would be valuable, so the researchers should delve deeper in this realm, especially in the conclusions.
  • The conclusions could be much stronger.  What are the applications of this research, and in what areas can this research influence resource management and policy?

Reviewer 2 Report

This is an article well structured with interesting findings. 

In section 2.2.2 it is not clear if the ten farmworkers were interviewed in each of the three original farms or in total. Please clarify.

Please include a section on farm dwellers' abilities to access land

Please add a section on game farming in Eastern Cape, current trends, impacts on social relationships. This should be also found in the conclusion section.

Reviewer 3 Report

I think this a strong case study, with an interesting theoretical framing. But there are a few major pieces I would like to see considered, which I highlight here in addition to some specific edits.

  1. Regime shifts vs adaptive cycle iterations

From a framing perspective, I don’t necessarily disagree with the idea a regime shift happened entirely, but I would like to see some more consideration about the difference between a regime shift and an iteration of the adaptive cycle – to me, this is an example of reorganization into an exploitation/conservation phase within the same adaptive cycle, not a reorganization that spins off into a new adaptive cycle. But you’re not super clear on whether that iteration would count as a regime shift. The key piece here in your definition is reversibility – to me, if you took down the fences and allowed free access to employees again, you’d probably end up with livestock grazing and the same degraded state. So again – I’m not necessarily disagreeing with the concept but would like to see the ‘regime shift’ considered in relation to the adaptive cycle. A point that might help explore the difference between a regime shift and a loop of the adaptive cycle is some temporal consideration of when the regime shift occurred (i.e. is there a distinct threshold between the two regimes or is it a drawn out process?).

Overall, I’d also like to see a bit more depth of reflection towards the end of the discussion or expanding what’s in the conclusion about the success and limitations of a regime shift framing.

  1. Remote sensing

Another major issue for me is the categorization of ‘newly revegetated’ land in the remote sensing analysis – I understand the value of pointing out this compared to bare land, but surely this overlaps with grassland (as does grazing land)? For example, on line 313 you talk about the revegetation of bare land and a decrease in grassland, but in my knowledge of grasslands wouldn’t first succession be of grass species i.e. creating grassland? Therefore, you need a much clearer analysis of what format the land is revegetating to. This will also help with the above point, as it will inform whether this is a regime shift or just an adaptation to a previous state. I’m sure you could break grassland down into multiple categories – ‘grassland – grazed by livestock’, ‘grassland – newly vegetated’, for example. Furthermore, in the labels for Figure 2 you say ‘rangeland’ vs grassland – please be consistent. And does ‘built up areas’ mean settlements/urbanized?

  1. Livestock farming vs game farming

I will say that the distinction between grazing livestock and grazing ‘wild’ animals seems minimal to me – but perhaps this is just lacking a distinction. Were the livestock herded and the wild animals are not? Can you make this clarification somewhere?

Furthermore, the end of the results section discusses that livestock farming was difficult during the droughts, but you don’t suggest why this is any different for game farming? I would have thought the quality of pasture is just as important if you are selling the game animals. But is it because quality of animal isn’t as important for the game farming value chains as the livestock farming value chains (which I assume were for dairy as you mention milking?)?

Finally, has the shift actually increased profits for the landowners?

  1. Wellbeing

I don’t particularly like that the abstract (in line 13-14) makes it sounds as this is an ecosystem service and wellbeing study – you have no wellbeing data beyond perceptions, and it isn’t mentioned until the discussion, then barely. So I’d feel more comfortable if that wasn’t placed as such a central concept in the abstract.

You say on line 337 that you show that economic benefits lead to individual financial wellbeing – but is this for the landowners or just farm workers? Because your data doesn’t seem to show that explicitly.

  1. Sample size and methods section

The sample size for participatory mapping (n=10) seems small for the population, which you don’t actually define. Do you know how many farmworkers lived on the area of study previously/still do? what are the features of that original population that lead to so many of your sample being women? I.e. were women hired primarily in the old farms? I thought the methods section was a little weak, with some details missing:

Line 181: ‘groups of between 3-5’ but only 10 farmworkers overall… so it should be 2-5 or 3-4 surely?

Line 182: can you give more detail on the sketch drawing – are they annotating am existing map? Drawing a bird-eye view sketch?

I think you also need to add what in the focus group protocol enabled you to create Table 1 – it’s not clear at the moment.
